# Accurate genetic and environmental covariance estimation with composite likelihood in genome-wide association studies

**Boran Gao**[1,2], **Can Yang**[3], **Jin Liu**[4], **Xiang Zhou**[1,2]*

**1** Department of Biostatistics, University of Michigan, Ann Arbor, MI, United States of America, **2** Center for Statistical Genetics, University of Michigan, Ann Arbor, MI, United States of America, **3** Department of Mathematics, Hong Kong University of Science and Technology, Hong Kong, China, **4** Centre for Quantitative Medicine, Program in Health Services and Systems Research, Duke-NUS Medical School, Singapore

* xzhousph@umich.edu

**Data Availability Statement:** GWAS data for the Wellcome Trust Case Control Consortium (WTCCC) study are available on www.wtccc.org. uk. Genotypes from the 1,000 Genomes Project are

## Abstract

Genetic and environmental covariances between pairs of complex traits are important quantitative measurements that characterize their shared genetic and environmental architectures. Accurate estimation of genetic and environmental covariances in genome-wide association studies (GWASs) can help us identify common genetic and environmental factors associated with both traits and facilitate the investigation of their causal relationship. Genetic and environmental covariances are often modeled through multivariate linear mixed models. Existing algorithms for covariance estimation include the traditional restricted maximum likelihood (REML) method and the recent method of moments (MoM). Compared to REML, MoM approaches are computationally efficient and require only GWAS summary statistics. However, MoM approaches can be statistically inefficient, often yielding inaccurate covariance estimates. In addition, existing MoM approaches have so far focused on estimating genetic covariance and have largely ignored environmental covariance estimation. Here we introduce a new computational method, GECKO, for estimating both genetic and environmental covariances, that improves the estimation accuracy of MoM while keeping computation in check. GECKO is based on composite likelihood, relies on only summary statistics for scalable computation, provides accurate genetic and environmental covariance estimates across a range of scenarios, and can accommodate SNP annotation stratified covariance estimation. We illustrate the benefits of GECKO through simulations and applications on analyzing 22 traits from five large-scale GWASs. In the real data applications, GECKO identified 50 significant genetic covariances among analyzed trait pairs, resulting in a twofold power gain compared to the previous MoM method LDSC. In addition, GECKO identified 20 significant environmental covariances. The ability of GECKO to estimate environmental covariance in addition to genetic covariance helps us reveal strong positive correlation between the genetic and environmental covariance estimates across trait pairs, suggesting that common pathways may underlie the shared genetic and environmental architectures between traits.

available on http://www.internationalgenome.org/. Summary statistics could be downloaded from the website listed in S1 Table.

**Funding:** This research was supported by National Institutes of Health grant R01HG009124 and National Science Foundation grant DMS1712933, both awarded to XZ. CY is supported by the Hong Kong Research Grant Council [16307818, 16301419, 16308120]. The funders had no role in study design, data collection and analysis, decision to publish, or preparation of the manuscript.

**Competing interests:** The authors have declared that no competing interests exist.

## Author summary

Phenotypic covariance between pairs of traits can be partitioned into two components: genetic covariance and environmental covariance. Effective partitioning of phenotypic covariance and accurate estimation of the genetic and environmental covariances can help us understand the relationship between traits and shed light on the causal and mediation mechanisms underlying disease etiology. Here we present a new computational method, GECKO, for estimating genetic and environmental covariances using summary statistics from GWASs. GECKO improves covariance estimation accuracy upon previous methods and provides environmental covariance estimates in addition to genetic covariance estimates. We illustrate the benefits of GECKO through extensive simulations and applications to 22 traits collected from five GWASs.

## Introduction

Phenotypic covariance between pairs of traits describes how one trait varies with respect to another in the population. Phenotypic covariance can be naturally partitioned into two components: the genetic covariance, which represents the part of the phenotypic covariance explained by common genetic factors; and the environmental covariance, which represents the part of the phenotypic covariance explained by common environmental factors [1]. The genetic and environmental covariances serve as important quantifications for measuring the relative contribution of genetic and environmental factors to phenotypic covariance, thus representing a key for understanding the nature versus nurture debate. Estimating and partitioning the phenotypic covariance between pairs of traits can facilitate the identification of common genetic and environmental factors underlying correlated traits, help investigate the potential causal relationship among them, and enhance our understanding of trait co-evolution under common evolutionary constraints [2–4].

A standard statistical model for estimating genetic and environmental covariances in genome-wide association studies (GWASs) is the multivariate linear mixed model (mvLMM) [5]. mvLMM extends the univariate linear mixed model commonly applied in genetic studies to accommodate multiple correlated phenotypes [5]. mvLMM relies on random effects to effectively capture the shared genetic and environmental architecture underlying multiple correlated traits. mvLMM has been widely applied for genetic covariance estimation in both animal breeding studies [6] and GWASs [3,5,7–9]. Standard fitting algorithms for mvLMM include the traditional restricted maximum likelihood estimation method (REML) and the recent method of moments (MoM). Compared to the traditional REML [3,5], the recent MoM approaches, including LD score regression (LDSC) [10] and GNOVA [11], require only summary statistics and are computationally efficient. Consequently, MoM algorithms have enabled applications of mvLMM for genetic covariance estimation in many large scale GWASs, revealing important genetic covariance structures underlying correlated traits.

Despite the popularity of MoM algorithms for covariance estimation, however, two important limitations of MoM exist. First, MoM algorithms are often statistically inefficient and produce less accurate parameter estimates when compared to the likelihood-based REML approach [12]. Indeed, estimation accuracy of MoM is known to vary depending on the underlying model parameters [12], and, as we show here, also depending on the sample composition of studies–whether the pair of phenotypes are measured on the same set of individuals or not. Second, and perhaps more importantly, existing MoM algorithms have been primarily focused

on genetic covariance estimation but have so far ignored environmental covariance estimation [10,11]. Failing to provide environmental covariance estimates is a significant restriction, considering that environmental covariance is an essential and indispensable component of the phenotypic covariance [1].

Here, we present a new method, for estimating both genetic and environmental covariances using GWAS summary statistics, that improves estimation accuracy of MoM algorithms while keeping computation in check. Our method relies on composite likelihood, is scalable computationally, uses only summary statistics, provides accurate genetic and environmental covariance estimates across a range of scenarios, and accommodates SNP annotation stratified covariance estimation. We refer to our method as Genetic and Environmental Covariance estimation by composite-likelihood Optimization (GECKO). We illustrate the benefits of GECKO with simulations and applications to 22 traits collected from five large scale GWASs.

## Methods

### Model specification

We consider an mvLMM [5] to model two phenotypes of interest jointly. These two phenotypes could be collected from a single study or from two studies with either non-overlapping or partially overlapping individuals. We denote $y_1$ and $y_2$ as the two phenotype vectors, measured on $n_1$ and $n_2$ individuals, respectively. We denote $n_s$ as the number of overlapping individuals who have both phenotypes measured. In the case of two separate studies, the sample size $n_1$ generally does not equal to $n_2$ and $n_s = 0$. In the case of a single study with completely overlapping individuals, the sample size $n_1 = n_2 = n_s$. Besides phenotypic measurements, all individuals have their genotypes measured on a common set of $m$ SNPs. We denote the $n_1$ by $m$ matrix $X_1$ as the genotype matrix for the $n_1$ individuals measured with the first phenotype and denote the $n_2$ by $m$ matrix $X_2$ as the genotype matrix for the $n_2$ individuals measured with the second phenotype. To facilitate computation, we center and standardize each phenotype vector as well as each column of the genotype matrices to have a mean of zero and a standard deviation of one. We consider the following regression model to related phenotypes to genotypes

$$
\begin{aligned}
y_1 &= X_1\beta_1 + \epsilon_1, \\
y_2 &= X_2\beta_2 + \epsilon_2,
\end{aligned}
\tag{1}
$$

where $\beta_1$ and $\beta_2$ are $m$-vectors of genotype effect sizes for the two phenotypes, respectively; and $\epsilon_1$ and $\epsilon_2$ are vectors of environmental effects/residual errors, with dimension $n_1$ and $n_2$, respectively.

We follow the standard mvLMM assumption and assume that the genetic effect sizes of $j$'th SNP on the two phenotypes follow a multivariate normal (MVN) distribution *a priori*

$$
\begin{pmatrix} \beta_{1j} \\ \beta_{2j} \end{pmatrix} \sim \text{MVN}\left( \begin{pmatrix} 0 \\ 0 \end{pmatrix}, \ \frac{1}{m} \begin{pmatrix} h_1^2 & \rho_g \\ \rho_g & h_2^2 \end{pmatrix} \right),
\tag{2}
$$

where $h_1^2$ represents the heritability of the first phenotype; $h_2^2$ represents the heritability of the second phenotype; and $\rho_g$ represents the genetic covariance, which characterizes the phenotypic covariance explained by genetic effects. We further define the genetic correlation as $\gamma_g = \frac{\rho_g}{\sqrt{h_1^2 h_2^2}}$.

For the $n_s$ individuals who have both phenotypes measured, we assume that their environmental effects for the $i$'th individual follow

$$\begin{pmatrix} \epsilon_{1i} \\ \epsilon_{2i} \end{pmatrix} \sim \text{MVN}\left( \begin{pmatrix} 0 \\ 0 \end{pmatrix}, \begin{pmatrix} 1 - h_1^2 & \rho_e \\ \rho_e & 1 - h_2^2 \end{pmatrix} \right), \tag{3}$$

where $\rho_e$ represents the environmental covariance, which characterizes the phenotypic covariance explained by environmental effects. The environmental correlation is further defined as $\gamma_e = \frac{\rho_e}{\sqrt{(1-h_1^2)(1-h_2^2)}}$. For the remaining individuals who only has one phenotype measured, we assume that $\epsilon_{1i}$ for the $i$'th individual follows a normal distribution $N(0, 1 - h_1^2)$ while $\epsilon_{2j}$ for the $j$'th individual follows another normal distribution $N(0, 1 - h_2^2)$.

We develop a composite likelihood based algorithm to perform estimation and inference for the mvLMM model defined in Eqs (1)–(3). The composite likelihood based algorithm requires only summary statistics as input and is both computationally and statistically efficient. For input summary statistics, we obtain the marginal z-score for each SNP-phenotype pair by fitting a univariate linear regression model. Specifically, the marginal z-score of $j$'th SNP for $d$'th phenotype ($d = 1, 2$) is $z_{dj} = X_{dj}^T y_d / \sqrt{n_d}$, where $X_{dj}$ represents $j$'th column of the genotype matrix $X_d$. We derive the marginal distribution for the two z-scores of each $j$'th SNP based on mvLMM. Such marginal distribution is in the following form (details in the S1 Text):

$$\begin{pmatrix} z_{1j} \\ z_{2j} \end{pmatrix} \sim MVN\left( \begin{pmatrix} 0 \\ 0 \end{pmatrix}, \frac{l_j}{m} \begin{pmatrix} h_1^2 n_1 & \rho_g \sqrt{n_1 n_2} \\ \rho_g \sqrt{n_1 n_2} & h_2^2 n_2 \end{pmatrix} + \begin{pmatrix} 1 & \frac{\rho n_s}{\sqrt{n_1 n_2}} \\ \frac{\rho n_s}{\sqrt{n_1 n_2}} & 1 \end{pmatrix} \right). \tag{4}$$

Here $\rho = \rho_e + \rho_g$; $l_j = \sum_{i=1}^m r_{ij}^2$ is the LD score for the $j$'th SNP computed based on a reference panel, with $r_{ij}$ being the correlation coefficient between $j$'th SNP and $i$'th SNP.

We denote $P(z_{1j}, z_{2j} | h_1^2, h_2^2, \rho_g, \rho)$ as the likelihood obtained based on Eq (4). The marginal z-scores for different SNPs are correlated with each other due to linkage disequilibrium (LD). Consequently, the joint likelihood for genome-wide marginal z-scores are in a complicated form, on which parameter estimation is computationally challenging to carry out. To enable estimation with summary statistics, we approximate the complicated joint likelihood with a relatively simple composite likelihood, which is represented as a weighted product of individual marginal likelihood across genome-wide SNPs [13]:

$$P(z_{11}, z_{21}, \dots, z_{1m}, z_{2m} | h_1^2, h_2^2, \rho_g, \rho) = \prod_{j=1}^m P(z_{1j}, z_{2j} | h_1^2, h_2^2, \rho_g, \rho)^{w_j}, \tag{5}$$

where each individual likelihood is powered to $w_j$, a $j$'th SNP specific weight. The weight $w_j$ is critical for ensuring estimation accuracy of the composite likelihood based algorithm in the presence of LD [14]. Intuitively, if $j$'th SNP is in LD with many other SNPs, then the marginal likelihood of $(z_{1j}, z_{2j})$ contains similar information as the marginal likelihood of the other SNPs that are in LD with the $j$'th SNP. Consequently, we can choose a small $w_j$ to down-weight the $j$'th marginal likelihood. In contrast, if the $j$'th SNP is in LD with only a limited number of SNPs, then the marginal likelihood of $(z_{1j}, z_{2j})$ contains important information that cannot be replaced by that of the other SNPs. Consequently, we want to choose a large $w_j$ to up-weight the $j$'th marginal likelihood. To captivate such intuition, we set $w_j = 1/l_j$ with $l_j$ being the LD score of the $j$'th SNP. Such choice of $w_j$ is optimum for achieving maximum estimation accuracy of composite likelihood for the specific setting where the SNP correlation matrix is block diagonal [14]: SNPs within blocks have identical genotypes with correlations among them

being one, while SNPs between blocks have zero correlations. In this setting, it has been shown that the optimal weighting choice is $w_j = 1/(m_i-1)$, where $m_i$ is the number of SNPs in $i$'th block to which the $j$'th SNP belongs [14]. This optimal weight choice $w_j = 1/(m_i-1)$ is equivalent to $w_j = 1/l_j$, the weighting choice we make for our method. In addition, the composite likelihood under our weighting choice of $w_j = 1/l_j$ is equivalent to the full likelihood when SNPs are uncorrelated with each other. The weight choice of $w_j = 1/l_j$ also relates our method to LDSC and MQS in the single phenotype setting (details in S1 Text) [12]. Therefore, our method can also be viewed as a natural extension of LDSC and MQS towards modeling multiple phenotypes.

With the above composite likelihood, we develop an expectation maximization (EM) algorithm [15] paired with an Newton-Raphson (NR) algorithm [16,17] to perform parameter estimation. In addition, we use the jackknife algorithm [10,18] to estimate standard errors for the parameter estimates. The composite likelihood-based algorithm allows us to perform unbiased parameter estimation [13] and make inference using only summary statistics in a computationally scalable fashion. The composite likelihood-based algorithm can also be easily extended to handle multiple genetic components in the presence of SNP annotations (details in the S1 Text). We refer to our method as genetic and environmental covariance estimate by composite-likelihood optimization (GECKO). GECKO is implemented as an R package, freely available at www.xzlab.org/software.html.

## Simulation settings

We used genotype data from the Welcome Trust Case Control Consortium (WTCCC) study and simulated phenotypes. Specifically, we obtained 384,733 SNPs from 2,938 control samples in the WTCCC. We filtered out SNPs with a minor allele frequency less than 5%, with a Hardy–Weinberg equilibrium p-value less than 0.001, or that are strand-ambiguous. We focused on the remaining 289,994 SNPs and simulated pairs of traits. In each simulation, we first simulated the genetic effect sizes for all SNPs based on Eq (2) with different genetic covariance values (more details below). We simulated the environmental effects based on Eq (3) with different environmental covariance values. We then summed the genetic effects and the environmental effects to obtain simulated phenotypes based on Eq (1). Afterwards, we computed marginal z-scores for each trait SNP pair using linear regression. We also computed LD scores using all individuals, with a window size set to be 1MB to minimize the influence of LD decay following the recommendation of LDSC [19]. Finally, we applied different methods to estimate the genetic and environmental covariance based on the resulting z-scores and LD scores. In the simulations, we examined different combinations of genetic and environmental covariance values, examined the influence of sample overlap on estimation accuracy, and examined the setting where genetic effects vary across different genetic annotations. In total, we examined four main simulation settings as described below:

**Setting I.** We kept the environmental covariance constant and varied the genetic covariance to evaluate the influence of environmental covariance on the estimation accuracy of the genetic covariance. Specifically, we set the environmental covariance to either -0.1, 0, or 0.1. For each environmental covariance value, we varied the genetic covariance from -0.4 to 0.4 by a step of 0.05. We also set the heritability of both traits to be 0.5. A total of 51 scenarios (3 environmental covariance x 17 genetic covariance values) were examined in setting I.

**Setting II.** We kept the genetic covariance constant and varied the environmental covariance to evaluate the influence of genetic covariance on the estimation accuracy of the environmental covariance. To do so, we set the genetic covariance to be either -0.1, 0, or 0.1. For each genetic covariance value, we varied the environmental covariance from -0.4 to 0.4 by a step of

0.05. We also set the heritability of both traits to be 0.5. A total of 51 scenarios (3 genetic covariance x 17 environmental covariance values) were examined in setting II.

**Setting III.**   Both settings I and II are based on the one study design where the two traits are measured on the same set of individuals. Here, we examined the influence of sample composition/overlap between the two traits on the estimation accuracy of the genetic covariance and environmental covariance. Specifically, we examined three study designs: the one study design where both traits are measured on the same set of individuals; the two study design where the two traits are measured on two sets of non-overlapping individuals; and the two partially overlapping study design where the two traits are measured on two studies with sample overlap. In the one study design, we used all 2,938 individuals to simulate both phenotypes. In the two study design, we used a randomly selected 1,500 individuals as study one sample to simulate the first phenotype and used the remaining 1,438 individuals as study two sample to simulate the second phenotype. In the two partially overlapping study design, we randomly selected 500 individuals from each study on top of the two study design and added them to the other study. Therefore, we have 2,000 individuals in study one and 1,938 individuals in study two, with 1,000 overlapping individuals. In each case, we set the heritability of both traits to be 0.5. We varied the genetic covariance from -0.4 to 0.4 by a step of 0.05. In the one study design and the two overlapping study design, we set the genetic covariance to be 0.1, and varied the environmental covariance from -0.4 to 0.4 by a step of 0.05. A total of 85 scenarios (3 study designs x 1 genetic covariance x 17 environmental covariance values + 2 study designs x 1 environmental covariance x 17 genetic covariance values) were examined in setting III.

**Setting IV.**   We examined genetic and environmental covariance estimation accuracy in the presence of multiple genetic covariance and environmental covariance. To do so, we divided SNPs into two categories following [20]: one functional category (132,557 SNPs) that includes all SNPs inside coding, UTR, promoter, exon or intron regions; and another non-functional category (157,437 SNPs) that include the remaining SNPs. We set the heritability of both traits to be 0.5 and evenly divided the heritability explained by each category of SNPs to be 0.25. For genetic covariance estimation, we set the genetic covariance of the non-functional annotation regions to be zero and varied the genetic covariance of the functional annotation regions from -0.2 to 0.2 by 0.025 following [11]. For environmental covariance estimation, we set the heritability of both traits to be 0.5 and evenly divided the heritability explained by each category of SNPs to be 0.25. In addition, we set the genetic covariance for the functional annotation region to be 0.05, set the genetic covariance for the non-functional annotation region to be 0, and varied the environmental covariance from -0.4 to 0.4 by a step of 0.05. A total of 85 scenarios (3 study designs x 1 genetic covariance x 17 environmental covariance values + 2 study designs x 1 environmental covariance x 17 genetic covariance values) were examined in setting IV.

Besides the above four main simulation settings, we also considered three additional simulation settings:

The **dense genotype setting** where we used 1,819,851 imputed WTCCC SNPs based on the 1,000 Genome phase three reference panel for simulations. We examined all 85 scenarios in the simulation setting III. The **moderate heritability setting** where we used a heritability of 0.15 instead of 0.5 and examined all 85 scenarios in simulation setting III. Note that the heritability of 0.15 is close to the mean heritability estimate in our real data application (mean = 0.155). The **mismatched LD setting** where the LD score is not computed from the data at hand but from a separate reference panel. Specifically, we used LD score computed from 503 individuals with European ancestry from the 1,000 Genomes project reference panel instead of that computed based on WTCCC. We considered mismatched LD setting in all 85 scenarios in setting III. The **overlap sample number misspecification setting** where we

examined the two partially overlapping study design in simulation setting III and set $n_s$ incorrectly to be 250, 500, 1,250, or 1,500 (while the true number is 1,000). We examined a total of 136 scenarios (4 $n_s$ choices x 1 genetic covariance x 17 environmental covariance values + 4 $n_s$ choices x 1 environmental covariance x 17 genetic covariance values) in this setting.

For each scenario in the simulation settings III and IV, we performed 1,000 simulation replicates. For each scenario in the remaining six simulation settings, we performed 100 simulation replicates. We calculated type I error and power based on these replicates to check the performance of GECKO under different sample compositions. Because different methods have different control of type I error, we compared the power of different methods at a fixed type I error rate instead of a nominal p-value threshold. Specifically, we ranked p-values from different methods under the null, obtained for each method its p-value threshold that corresponds to a 5% type I error rate, and used this p-value threshold for the given method as the cutoff to calculate its power. Therefore, a different p-value threshold is used for each different method, ensuring fair power comparison at a fixed type I error rate.

## Compared methods

We mainly compared our method with two existing summary statistic based methods: LDSC and GNOVA. Both these two methods estimate the genetic covariance through MoM and both of them rely on marginal z-scores and LD scores as input. In LDSC, we used the default setting for genetic covariance estimation. In GNOVA, we also used the default setting in the one study design and the two partially overlapped study design; and we followed the recommendation of GNOVA in the two separate study design to improve estimation performance. Neither GNOVA nor LDSC estimate can directly estimate or test environmental covariance. LDSC outputs an estimate for the intercept covariance, which can be approximately considered as the summation of the genetic covariance and the environmental covariance. Therefore, we *post hoc* processed the LDSC output, obtained the difference between the intercept covariance estimate and the genetic covariance estimate and treated it as an *ad hoc* estimate for the environmental covariance. However, we are unable to test for the environmental covariance using LDSC: LDSC does not output the standard error for the environmental covariance and it is not straightforward to extract such standard error in a *post hoc* fashion. For GNOVA, we also obtained the LDSC estimate of the LDSC intercept covariance, from which we subtracted the GNOVA genetic covariance estimate to obtain an *ad hoc* estimate for the environmental covariance. In stratified analysis, we obtained the GNOVA genetic covariance estimates through the above procedure in each annotation region separately. Afterwards, we obtained the environmental covariance estimate by taking the difference between the LDSC intercept estimate and the summation of GNOVA genetic covariance estimates across annotation regions. Besides the above summary statistics based methods, we also compared to the individual level data based method mvLMM. Specifically, we fitted mvLMM using GCTA and examined the simulation setting III with different sample overlap compositions. Finally, we note that we used $1/l_j$ as the weight for the composite likelihood in GECKO throughout the text as explained above, where $l_j$ is the LD score of the $j$'th SNP.

## Real data analysis

We applied our method to analyze GWAS summary statistics for 22 traits collected from five GWASs. These 22 traits include nine non-impedance traits from the UK biobank [21] that have SNP heritability greater than 0.05, four lipid traits from the Global Lipids Genetics Consortium (GLGC) [22], five birth traits from Early Growth Genetics (EGG) Consortium [23–28], two anthropometric measures from the Genetic Investigation of ANthropometric Traits

(GIANT) Consortium [29,30], and two bone density measurements from the GEnetic Factors for OSteoporosis Consortium (GEFOS) [31]. Details of these traits are provided in the S1 Table. The 22 traits can be classified into four categories: anthropometric traits (e.g., height and BMI), hematological traits (e.g., WBC and RBC), birth traits (e.g., birth weight and birth length), and metabolic traits (e.g., TG and HDL). For analysis, we first obtained a common set of SNPs shared across all traits. We filtered out SNPs with minor allele frequency (MAF) less than 0.05, with Hardy-Weinberg equilibrium (HWE) p-value less than 0.001, or that are strand-ambiguous. We used the R package liftOver to match SNP base positions onto NCBI 37 build whenever necessary. We further overlapped SNPs with those available in the 1,000 Genomes project and focused on a final set of 611,444 SNPs for analysis. On the final set of SNPs, we calculated LD scores based on 503 individuals with European ancestry from the 1,000 Genomes project [32] with a window size of 1 MB as recommended by LDSC [19]. We applied GECKO together with LDSC and GNOVA to analyze all trait pairs among the 22 traits. We declared significance in covariance estimates based on Bonferroni corrected thresholds. For each trait pair in turn, we also performed gene set enrichment analysis using GSA-SNP2 [33] to identify enriched gene sets. We further compared the proportion of enriched gene sets in trait pairs with either significant or non-significant genetic covariances.

In addition, we examined the performance of different methods on denser genotypes in the real data. Specifically, we focused on the 13 phenotypes from GLGC (4) and UKBB (9) that have a much higher number of genotypes as compared to the remaining phenotypes. We obtained a common set of SNPs among these 13 phenotypes and retained SNPs that are also available in the 1,000 Genomes project reference panel. We then filtered out SNPs with minor allele frequency (MAF) less than 0.05, with Hardy-Weinberg equilibrium (HWE) p-value less than 0.001, or that are strand-ambiguous. After filtering, we obtained a final set of 1,731,189 SNPs for analysis.

## Results

GECKO is described in the Methods section, with technical details provided in the S1 Text. Briefly, GECKO analyzes pairs of traits one at a time. For each pair, it relies on a composite likelihood-based algorithm to estimate and test the genetic and environmental covariances.

### Genetic and environmental covariance estimation

We performed simulations to compare the estimation accuracy of GECKO with two existing approaches, LDSC and GNOVA. Simulation and comparison details are provided in the Methods. Briefly, we used genotypes from WTCCC to simulate pairs of traits. We examined a total of 238 simulation scenarios in four main simulation settings. We first examined the one study design where both traits are measured on the same set of individuals (n = 2,938). We examined the estimation accuracy of the three methods for a range of genetic covariance values in the setting where the environmental covariance is either zero (Fig 1A and 1B), positive (S1A and S1B Fig), or negative (S1C and S1D Fig). All three methods provide approximately unbiased estimates of the genetic covariance regardless of the environmental covariance values (Figs 1A and S1A and S1C). To quantify estimation accuracy, we computed the mean squared error (MSE) for estimates obtained from different methods. To facilitate visualization, we also contrasted the MSE from each of the other two methods to GECKO by computing an MSE ratio (Figs 1B and S1B and S1D). The MSE ratio measures the inverse of the relative statistical efficiency of a given method with respect to GECKO: an MSE ratio above one suggests that GECKO provides more accurate estimates while an MSE ratio below one suggests the opposite. In addition, the MSE ratio directly quantifies how effective the two methods are in using

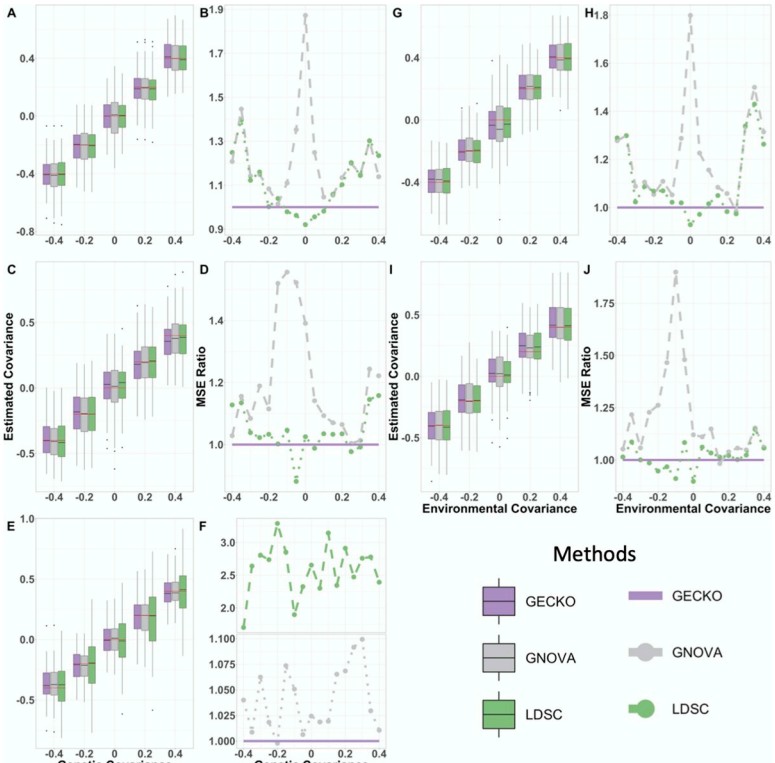

**Fig 1. Comparison of genetic and environmental covariance estimation of different methods in simulations.**
Compared methods include GECKO (purple), GNOVA (grey), and LDSC (green). Results are shown for the one study design (first row: A, B, G, H), two partially overlapped study design (second row: C, D, I, J), and two separate study design (third row: E, F). Boxplots display estimated genetic covariances (A, C, E) and environmental covariances (G, I) on y-axis versus the true covariances on x-axis across simulation replicates. Estimation accuracy is measured by the ratio of mean square errors (MSE), which contrast the MSE from GNOVA or LDSC with respect to GECKO, across various true covariances on x-axis, for genetic (B, D, F) and environmental covariances (H, J). An MSE ratio below one suggests that GECKO performs worse than the other method; above one otherwise.

the samples, as the estimation variance is a function of sample size due to the central limit theorem. For example, an MSE ratio of 1.11 would imply that GECKO effectively increases the sample size by 1.11−1 = 11% as compared to the other method. The MSE ratio results show that GECKO provides more accurate genetic covariance estimates than the other two methods across a wide range of scenarios. The performance of GECKO is often followed by LDSC. For example, when the environmental covariance is zero, the MSE ratio of LDSC ranges from 0.92 to 1.39 with a mean of 1.11 (with the ratio in 70.6% of simulation replicates being above 1), while the MSE ratio of GNOVA ranges from 1.01 to 1.87 with a mean of 1.22. The overall results do not vary much with respect to the environmental covariance values. For example, when the environmental covariance is 0.1, the MSE ratio of LDSC ranges from 0.86 to 1.48 with a mean of 1.16 (with 82.3% above 1), while the MSE ratio of GNOVA ranges from 1.04 to 1.69 with a mean of 1.26. Similarly, when the environmental covariance is -0.1, the MSE ratio of LDSC ranges from 0.95 to 1.59 with a mean of 1.13 (with 76.5% above 1), while the MSE ratio of GNOVA ranges from 1.05 to 1.65 with a mean of 1.25. The estimation accuracy of the genetic correlation with varying environmental correlation showed a similar overall pattern as the results on genetic covariance estimation (S2A, S2B and S3A–S3D Figs). Besides genetic covariance estimation, we also examined the estimation accuracy of environmental covariance with varying genetic covariances. Because LDSC and GNOVA do not provide environmental

covariance estimates, we had to post-process LDSC and GNOVA output to obtain these estimates in an *ad hoc* fashion (details in Methods). The results on environmental covariance estimation are largely consistent with genetic covariance estimation results. For example, all three methods provide reasonably unbiased environmental covariance estimates regardless of the genetic covariance values (Figs 1G and S1E and S1G). GECKO produces more accurate environmental covariance estimates than both LDSC and GNOVA in most (78.4%) simulation scenarios (Figs 1H and S1F and S1H).

Examining the detailed scenarios in which different methods perform well yields further insights. Specifically, LDSC is particularly effective in the scenario where the genetic and environmental covariance sums to zero. For example, when the environmental covariance is zero, the estimate by LDSC is accurate when the true genetic covariance is close to zero, even slightly more so than GECKO. However, the performance of LDSC degrades quickly when the summation of genetic and environmental covariance differs far away from zero. The performance dependence of LDSC with respect to the summation of the true covariances closely resembles its performance dependence on the true heritability for heritability estimation–LDSC heritability estimate is also more accurate when the true heritability is close to zero [12]. In contrast, the performance of GNOVA is almost in the opposite direction of LDSC: the estimation accuracy of GNOVA is the worst when the genetic and environmental covariance sum to zero and improves when the sum deviates away from zero. The different performance dependence on the true covariance values in GNOVA and LDSC is likely resulted from different MoM type algorithms used in the two methods [12]. Different from LDSC and GNOVA, the performance of GECKO is reasonably stable across a range of genetic and environmental covariance values (Figs 1B and S1B and S1D), supporting the benefits of performing inference in a composite likelihood framework.

## Sample overlap, type I error control, and stratified analysis

Besides the above settings where both traits are measured on the same set of individuals, we also examined settings where the two traits are measured on two studies, with either no overlapping or partially overlapping individuals between the two. Specifically, we set the first study sample size $n_1 = 1,500$ and the second study sample size $n_2 = 1,438$ in the two separate study design. We also set $n_1 = 2,000$, $n_2 = 1,938$, with overlap sample size $n_s = 1,000$ in the two partially overlapped study design. The results are largely consistent with the one study design described in the previous section. For example, all methods provide approximately unbiased estimates for both the genetic covariance (Fig 1C and 1E) and the environmental covariance (Fig 1G and 1I). Compared to the other two methods, GECKO produces more accurate estimates of the genetic covariance in most (82.3%) scenarios (Fig 1D and 1F) and more accurate estimates of the environmental covariance in most (70.6%) scenarios (Fig 1H and 1J). As expected, the proportion of individual overlap between studies positively influences the estimation accuracy of all methods. For example, the average MSE of the genetic covariance estimates by GECKO increases from 0.012 in the one study design to 0.026 in the two partially overlapped study design, and further to 0.043 in the two separate study design. Certainly, all these compared methods are based on summary statistics and use either method of moments or composite likelihood. Consequently, these methods produce less accurate genetic and environmental covariance estimates as compared to the full likelihood based method mvLMM, which uses individual level data (S4 Fig). For example, the MSE ratio of mvLMM over GECKO for genetic covariance estimation is much smaller than one in the presence of sample overlap though it becomes closer to one in the two separate study design. Overall, the results suggest that the performance of GECKO is robust with respect to sample compositions.

**Table 1. Type I error control of different methods under null simulations.**

| Type I Error (alpha = 0.05, 0.005) | Genetic Covariance | | | Environmental Covariance | |
|---|---|---|---|---|---|
| | One Study | Two Study | Overlapped | One Study | Overlapped |
| GECKO | 2% (0.2%) | 2% (0.8%) | 2% (0.6%) | 2% (0.2%) | 3% (0.5%) |
| LDSC | 3% (0.3%) | 1% (0) | 2% (0.1%) | NA | NA |
| GNOVA | 37% (17.6%) | 7% (0.8%) | 27% (10.9%) | NA | NA |

The table displays the type I error of different methods at a nominal p-value threshold of 0.05 and 0.005 (in parentheses) for testing the genetic covariance (first three columns) or the environmental covariance (last two columns) under null simulations with three different study designs. The three different study designs include the one study design, two separate study design, and two study design with partially overlapped individuals. LDSC and GNOVA cannot be used for testing the environmental covariance so NAs are shown in the corresponding cells.

Besides estimation, we found that both LDSC and GECKO control for type I error well for genetic covariance testing and produce conservative p-values across study designs (Table 1). Consistent with [11], GNOVA also produces reasonably calibrated type I error control in the two separate study design. However, GNOVA produces inflated type I error in both one study design and two overlapping study design, sometimes several times larger than expected (Table 1); both these two study designs were not explored in [11]. In addition, GECKO is the only method capable of testing the environment covariance, producing controlled type I error and conservative p-values across settings (Fig 2D and 2E and Table 1). Because different methods have different control of type I error, we compared the power of different methods at a fixed type I error rate instead of a nominal p-value threshold. In the analysis, we found that GECKO is also more powerful than the other two methods in testing genetic covariance, despite the relatively small sample sizes used in simulations. For example, in the one study design, GECKO improves power by 4.05% and 32.4% on top of LDSC and GNOVA, respectively (Fig 2A). In the partially overlapped study design, GECKO improves power by 3.72% and 9.44% on top of LDSC and GNOVA, respectively (Fig 2C). GNOVA is even more powerful than the other two methods in the two separate study design (Fig 2E).

Next, we examined the performance of GECKO in estimating annotation-stratified genetic covariance and environmental covariance. To do so, we partitioned the whole genome into two non-overlapping annotation categories and applied both GECKO and GNOVA to estimate the two annotation-stratified genetic covariances. We did not include LDSC here as it cannot estimate annotation-stratified covariances. The results in the annotation-stratified analysis are largely similar to the results for non-stratified analysis. For example, both GECKO and GNOVA provided approximately unbiased estimates of the partitioned genetic covariances regardless of the study design (Fig 3A, 3C and 3E). GECKO also produces more accurate genetic covariance estimates than GNOVA. For example, in the one study design, the MSE ratio of GNOVA ranges from 1.01 to 1.08 with a mean of 1.05 (Fig 3B). The relative accuracy of GECKO with respect to GNOVA improves with reduced sample overlap. For example, the MSE ratio of GNOVA to GECKO ranges from 1.10 to 1.23 with a mean of 1.17 in the two separate study design (Fig 3D), and ranges from 1.16 to 1.44 with a mean of 1.36 in the two partially overlapping study design (Fig 3F). In terms of environmental covariance, GNOVA cannot directly perform estimation in the presence of stratified annotations so we had to use a *post hoc* procedure to obtain environmental covariance estimates from GNOVA. Consistent with results on the unstratified environmental covariance estimation, GECKO provided approximately unbiased estimate of environmental covariance regardless of study design (Fig 3G and 3I) and is more accurate than GNOVA (Fig 3H and 3J).

Besides estimation, we found that GECKO is more powerful than GNOVA in testing genetic covariance in the one study design and the two partially overlapped study design. For

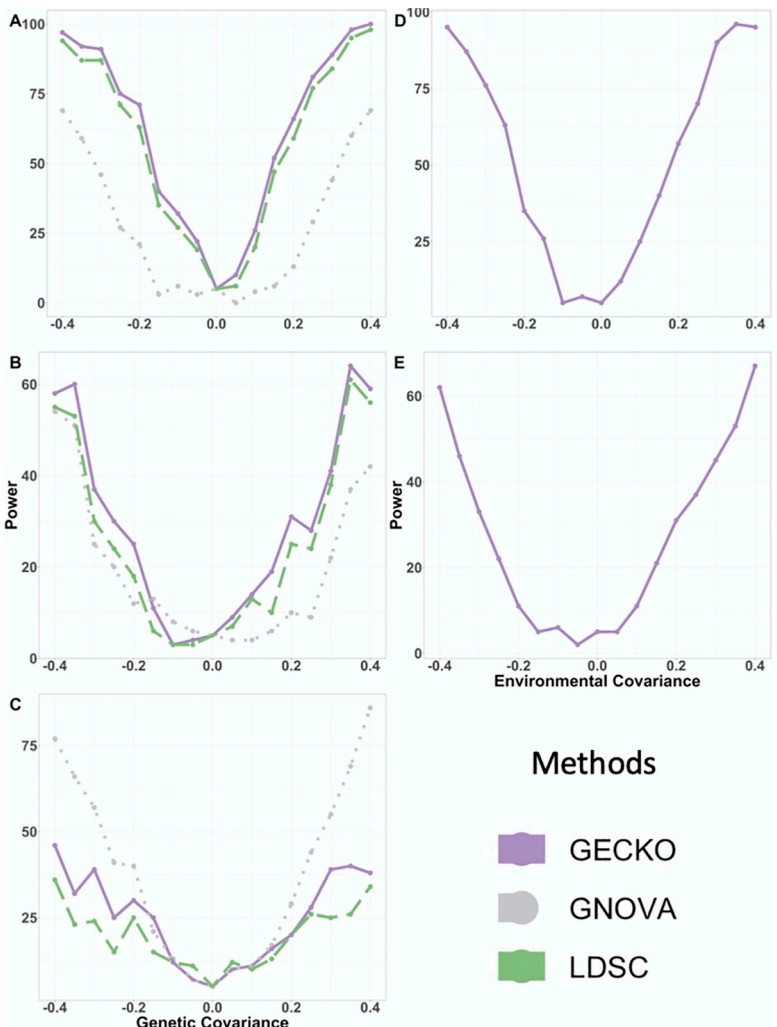

**Fig 2. Power comparison of different methods in detecting non-zero genetic and environmental covariances in simulations.** Compared methods include GECKO (purple, solid line), GNOVA (grey, dotted line), and LDSC (green, dashed line). Results are shown for the one study design (first row: A, D), two partially overlapped study design (second row: B, E) and two separate study design (third row: C). Power (y-axis) of different methods are shown with respect to the true covariances (x-axis) for detecting non-zero genetic covariances (A, B, C) and environmental covariances (D, E). Power is shown based on a type I error of 0.05 but not a nominal p-value of 0.05. The power of GNOVA and LDSC for detecting non-zero environmental covariance are not shown because GNOVA and LDSC cannot test for environmental covariance.

example, in the one study design, GECKO improves power by 17.94% on top of GNOVA (S5A Fig). In the partially overlapped study design, GECKO improves power by 2.59% on top of GNOVA (S5B Fig). However, GNOVA is more powerful than GECKO in the two separate study design (S5C Fig), presumably due to its implicit modeling assumption that the environmental covariance is zero, which happens to be true in the two separate study design.

## Alternative simulations and model misspecifications

Besides the main simulations, we examined the performance of GECKO and the other methods with additional simulation settings including those under different model misspecifications. We first examined the performance of different methods under simulations with denser

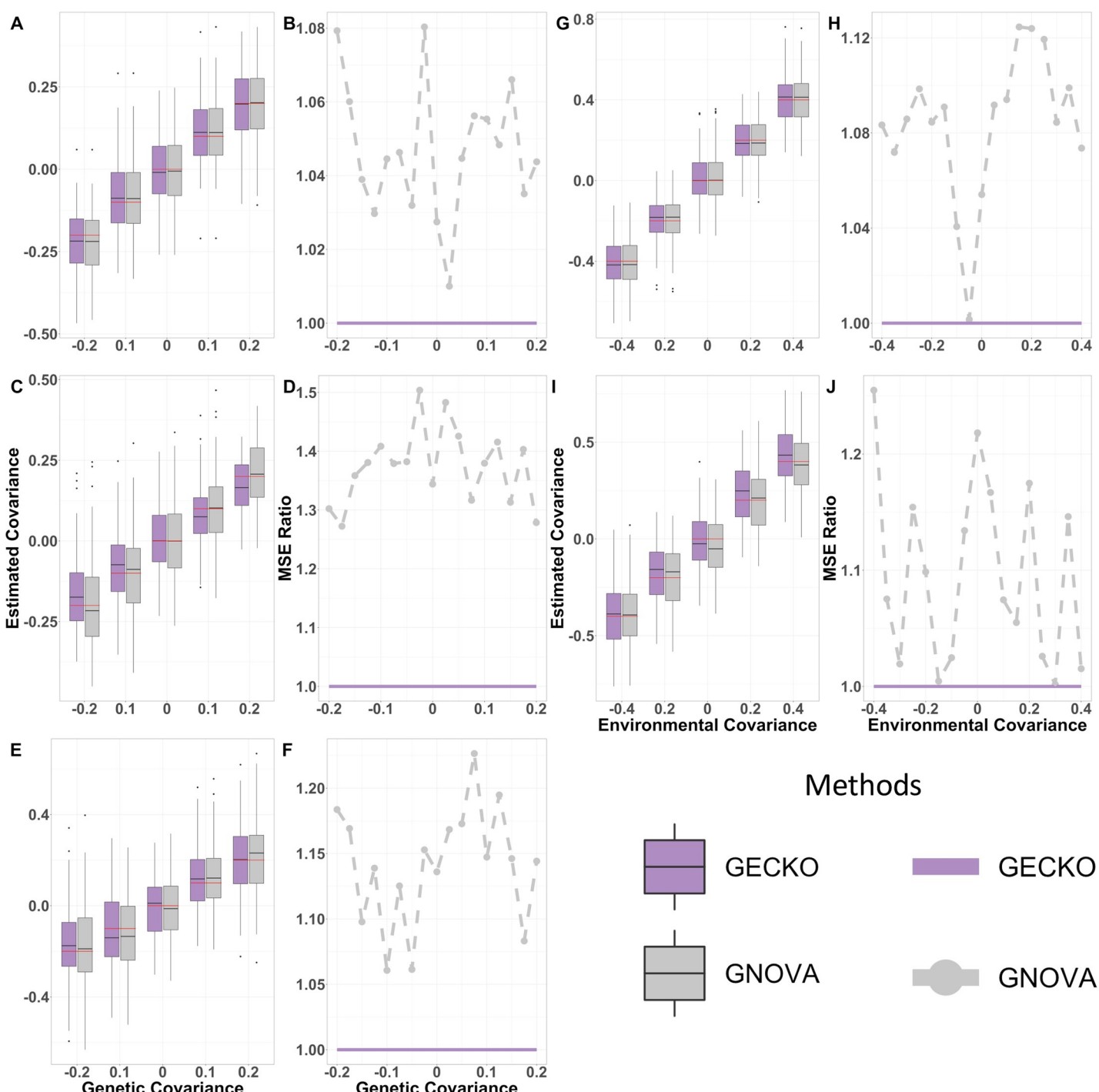

**Fig 3. Comparison of genetic and environmental covariance estimation in presence of stratified genetic components for different methods in simulations.**
Compared methods include GECKO (purple) and GNOVA (grey). Results are obtained in the presence of stratified genetic components. Results are shown for the one study design (first row: A, B, G, H), two partially overlapped study design (second row: C, D, I, J) and two separate study design (third row: E, F). Boxplots display the estimated total genetic covariances (A, C, E) and environmental covariances (G, I) on y-axis versus the true covariances on x-axis across simulation replicates. Estimation accuracy is measured by the ratio of mean square errors (MSE), which contrast the MSE from GNOVA or LDSC with respect to GECKO, across various true covariances on x-axis, for genetic (B, D, F) and environmental covariances (H, J). An MSE ratio below one suggests that GECKO performs worse than the other method; above one otherwise.

genotypes. Specifically, we used imputed genotypes and performed simulations under setting III. We found that the results under denser genotypes are largely consistent with the above main results (S6 Fig). We then examined the performance of different methods based on a moderate heritability of 0.15 instead of 0.5. A heritability of 0.15 is close to the mean heritability estimate obtained in our real data application. The results with moderate heritability are also largely consistent with the main results (S7 Fig). We further examined a LD mismatch setting, where we calculated LD scores and subsequent composite likelihood weights based on the 1,000 Genomes project reference panel instead of directly from the data at hand. The results are largely consistent with the main simulations: all methods provide approximately unbiased estimates for both the genetic (S8A, S8C and S8E Fig) and environmental covariances (S8G and S8I Fig) while GECKO produces more accurate estimates of the genetic covariance (S8B, S8D and S8F Fig) and the environmental covariance in most (82.3% and 76.5%) scenarios (S8H and S8J Fig). We also examined another model misspecification setting where the number of overlapping individuals is mis-specified. Because the genetic covariance and the number of overlapping individuals are two parameters that are separable from each other in the model, the estimation of genetic covariance from different methods would not be affected by the misspecification of the number of overlapping individuals (S9A, S9C, S9G, and S9I Fig). Indeed, the MSE ratio comparing different methods remains the same (S9B, S9D, S9H and S9J vs S9F Fig). However, the environmental covariance and the number of overlapped individuals are two parameters that are coupled together in the model and are thus not identifiable from each other. Consequently, when the number of overlapping individuals is mis-specified, the environmental covariance estimate would become biased for all three methods (S10A, S10C, S10G and S10I Fig), although the MSE ratio comparing different methods remains largely the same (S10B, S10D, S10H and S10J vs S10F Fig).

Finally, we note that the computational complexity of GECKO scales linearly with the number of individuals and with the number of SNPs. While GECKO requires iterative optimization, the linear computational complexity of GECKO makes it reasonably efficient as compared to MoM based approaches. Indeed, across simulation settings, the average computing time for covariance estimation is 95.43s, 7.41s, and 7.67s for GECKO, LDSC and GNOVA, respectively.

## Real data applications

We applied GECKO and LDSC to analyze 22 quantitative traits from five different GWASs. These traits belong to five distinct phenotype groups that include birth traits, lipid traits, bone density traits, blood traits, and anthropometric measures. The heritability estimates of 22 traits range from 0.047 to 0.368 with a mean heritability estimate being 0.155. We focused on all 231 pairs of traits and a common set of 611,444 SNPs for analysis. We did not include GNOVA in comparison as it does not provide a well-controlled type I error for testing. Across pairs of traits, the genetic correlation estimates from GECKO range from -0.731 to 0.945, and the environmental correlation estimates from GECKO range from -0.363 to 0.875. In the analysis, GECKO identified 50 significant genetic correlations based on Bonferroni correction (p-value $< 0.00022$ (0.05/231); Fig 4A). Among them, a higher proportion of significant genetic correlation are obtained among trait pairs collected in the same study (19 out of 54; 35.19%) than that collected in separate studies (31 out of 177; 17.51%), consistent with the fact that traits are collected in the same study if they likely share a common genetic background. Consistent with the lower power of LDSC in simulations, LDSC identified 24 significant genetic correlations, the majority of which (19) were also identified by GECKO (S11 Fig) and 19 of which were obtained among trait pairs collected in the same study. Among the 19 pairs of traits with significant genetic correlation identified by both methods, the majority of them

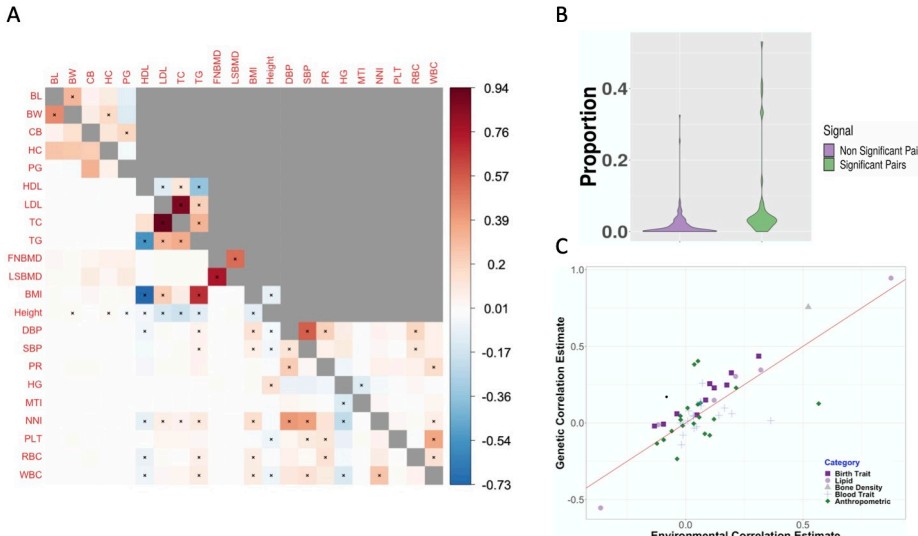

**Fig 4. Genetic and environmental correlation estimation by GECKO across pairs of 22 GWAS traits.** A: heatmap displays genetic (lower triangle) and environmental (upper triangle) correlation estimates for pairs of traits from GECKO. A cross in the square represents statistically significant correlation estimates for the trait pair after Bonferroni correction. Environmental correlation estimation is carried out only for pairs of traits that are collected in the same study (non-grey boxes). B: violin plot displays the proportion of significantly enriched gene sets detected for pairs of traits with non-significant genetic correlation estimates (purple) and for pairs of traits with significant genetic correlation estimates (green). C: scatterplot contrasts the genetic correlation estimates (y-axis) versus the environmental correlation estimates (x-axis) for pairs of traits that are collected in the same study. Traits are organized into five different phenotype categories (figure legend).

have previous literature support (S2 Table). For example, both methods identified positive genetic correlation between LSBMD and FNBMD, and between BMI and TG, as well as negative genetic correlation between BMI and HDL, and between TG and HDL [10,11]. In terms of the 31 significant genetic correlation obtained only by GECKO, many of them also have previous literature support. For example, the positive genetic correlation between BMI and WBC (correlation = 0.091; p-value = $2.24\times10^{-12}$) is consistent with the previous study that common inflammatory pathways may underlie both traits: high BMI and obesity is considered as a chronic inflammatory condition with which elevated WBC is widely recognized to be associated with [34,35]. As another example, the positive genetic correlation between BMI and SBP (correlation = 0.06; p-value = $1.46\times10^{-6}$) suggests that either common genetic pathways may underlie both BMI and SBP or that a potentially causal positive effect of BMI on hypertension exist as suggested previously [36]. Indeed, gene set enrichment analysis (GSEA) identified thyroid hormone receptor binding pathway (WBC q-value = 0.00079; BMI q-value = 0.033) and cytokine-mediated signaling pathway (WBC q-value = 0.0014; BMI q-value = 0.013) as enriched in genes associated with BMI or SBP [37–39]. Similarly, the positive genetic correlation between BMI and LDL (correlation = 0.23; p-value = $4.14\times10^{-9}$) suggests that either common genetic pathways underlie both BMI and LDL or BMI has a positive and potentially causal effect on LDL as suggested previously [40,41]. Indeed, GSEA identified lipid storage pathway (BMI q-value = 0.022; LDL q-value = $1.29\times10^{-7}$) and receptor biosynthetic process pathway (BMI q-value = 0.0067; LDL q-value = 0.0001) as enriched in genes associated with BMI or LDL, supporting common genetic regulation underlying the two traits. Besides positive genetic correlation, the identified negative genetic correlation between pairs of traits are also consistent with previous literature. For example, the negative genetic correlation identified between height and LDL (correlation = -0.16; p-value = $2.96\times10^{-13}$) is consistent with

previous studies that height associated SNPs display negative associations with LDL [42]. In addition, GSEA analysis revealed that, consistent with previous studies [43], the steroid bio-synthetic process is enriched in genes associated with both height and LDL (LDL q-value = 0.0002; height q-value = 0.038). Similarly, the negative genetic correlation between WBC and height (correlation = -0.065, p-value = $1.60 \times 10^{-12}$) is also supported by GSEA analysis, where we found that the receptor biosynthetic process (WBC q-value = 0.01, height q-value = 0.01) and the multicellular organism growth pathways (WBC q = 0.004, height q-value = 0.009) are both enriched in genes associated with either of the two traits [44]. Overall, the proportion of enriched pathways identified in trait pairs with a significant genetic correlation is 4.4 fold higher than that in trait pairs with non-significant genetic correlation (7.0% vs 1.6%; p-value $< 1 \times 10^{-4}$; Fig 4B).

Besides genetic correlation, we also applied GECKO to obtain environmental correlation estimates among 54 trait pairs in which both traits were measured on a common set of individuals. We did not apply LDSC for environmental correlation estimation becaues it cannot test the environmental correlation. The estimated environmental correlation estimates are positively correlated with the genetic correlation estimates across trait pairs (Spearman correlation = 0.707; Fig 4C), with the two estimates sharing the same sign for majority of trait pairs (45 out of 54; 83.3%). The positive correlation between the environmental correlation estimates and genetic correlation estimates is highest for pairs of traits in the lipid phenotype group (correlation = 0.98) where 100% of the trait pairs share the same sign, and is the lowest for pairs of traits in the blood phenotype group (correlation = 0.25) where 66.7% of the trait pairs share the same sign. 9 trait pairs (out of 54 pairs) showed opposite signs of the genetic and environmental covariance estimates, though none of these pairs were statistically significant. Among the 54 analyzed trait pairs, GECKO identifies 24 significant environmental correlation based on Bonferroni correction (p-value < 0.0009 (0.05/54); Fig 4A). Among them, 11 pairs have significant genetic correlation estimates with the same sign. The significant environmental correlation estimates are identified among pairs of traits within each of the three phenotypic categories that include birth traits, metabolic traits, and anthropometric traits (S3 Table). For example, significant positive environmental correlation is observed between birth length and birth weight (correlation = 0.31; p-value = $4.24 \times 10^{-37}$), between birth weight and infant head circumference (correlation = 0.18; p-value = $1.25 \times 10^{-5}$), and between childhood BMI and pubertal growth (correlation = 0.19; p-value = $3.49 \times 10^{-3}$). Such positive environmental correlation estimates suggest that common environmental factors, such as maternal health status and maternal dietary, may influence birth traits in a coherent fashion [45–48]. As a second example, the significantly positive environmental corelation estimates between bone mineral density at femoral neck (FNBMD) and lumbar spine (LSBMD; correlation = 0.52; p-value = $3.02 \times 10^{-63}$) suggest that environmental factors such as physical activity may influence bone mineral density at two different anatomical locations in a similar fashion [49]. As a third example, the significantly negative environmental correlation estimate between height and BMI (correlation = -0.09, p-value = $4.93 \times 10^{-25}$) suggests that environmental factors such as television watching, physical activity, and smoking habits may influence height and BMI in different directions [50–52]. Finally, paring the genetic correlation estimates with the environmental correlation estimates helps us better understand the main contribution of phenotypic correlation between pairs of traits. For example, previous studies have identified a positive phenotypic association between WBC and pulse rate [53]. Such positive phenotypic association appeared to be accounted for by a positive environmental correlation between the traits (genetic correlation = 0.097, p-value = $2.28 \times 10^{-3}$; while environmental correlation = 0.16, p-value = $6.90 \times 10^{-8}$), suggesting that common environmental factors for inducing inflammatory response could underlie heart rate variation as previous studies suggest [34,54].

Finally, we examined the influence of SNP density on covariance estimation. Specifically, we focused on 13 phentoypes that have a larger number of SNPs than the remaining phenotypes and analyzed the 78 trait pairs among them using the dense set of SNPs (details in Methods). In the analysis, GECKO identified 37 significant genetic correlations based on Bonferroni correction (p-value < 0.00064) while LDSC identified 24, 22 of which were also identified by GECKO. Importantly, the genetic correlation estimates obtained with the dense set of SNPs is highly consistent with the estimates obtained earlier (Spearman correlation = 0.865). In addition, 28 significant estimates are shared in common between the significant genetic correlations detected using the dense set of SNPs and the using early non-dense SNPs (S4 Table). Besides genetic corrections, the environmental correlation estimates obtained with the dense set of SNPs is highly correlated with the estimates obtained with the earlier non-dense set of SNPs, with Spearman correlation being 0.98 between the two. In addition, 15 significant environmental correlation estimates are shared in common between the significant environmental correlations detected using the dense set of SNPs and the early non-dense set of SNPs (S4 Table). Overall, SNP density does not appear to influence the GECKO results.

## Discussion

We have presented our method, GECKO, for estimating genetic and environmental covariances in GWASs. The same estimation formula of GECKO can be derived under either a random design genotype matrix assumption as assumed in LDSC or a fixed design genotype matrix assumption as assumed in GNOVA (S1 Text), highlighting the robustness of GECKO estimation under different modeling assumptions on the genotype matrix. GECKO provides accuracy genetic and environmental covariance estimates across a range of scenarios including various study sample compositions. By providing the option of estimating and testing the environmental covariance in additional to the genetic covariance, GECKO complements existing approaches for fitting mvLMM and facilities the investigation of both the genetic and environmental underpinning of complex traits. The composite-likelihood framework of GECKO is also general and may be paired with several recent methodological advances in the field for a variety of additional applications. For example, GECKO may be paired with the likelihood framework of the high definition likelihood inference (HDL) method and rely on block-wise LD matrix approximation to further improve estimation accuracy [55]. GECKO may be extended for local genetic correlation estimation based on the recent SUPERGNOVA framework by decorrelating summary statistics among local SNPs with the top principle components extracted from the local LD matrix [56]. GECKO may be extended towards trans-ethnic genetic correlation estimation based on a similar strategy used in [57]. Exploring extensions of GECKO for these additional applications may yield fruitful results in the future.

GECKO relies on composite likelihood for covariance estimation. A key ingredient of composite likelihood is the use of SNP weights, which weight the individual marginal likelihood of each SNP to achieve improved estimation accuracy. The optimal choice of SNP weights is only known in specific settings including the setting where LD displays a block-diagonal pattern and the setting where SNPs are all uncorrelated with each other. Our use of the inverse LD score as SNP weights corresponds to such optimal weights in the absence of LD or in the presence of block diagonal LD structure. Certainly, such SNP weight choice may not be optimal for general non-block diagonal LD structures. Consequently, future explorations on the use of other SNP weighting choices may lead to more accurate covariance estimates through the composite likelihood estimation framework.

We have primarily focused on estimating covariances for quantitative traits using mvLMM. Estimating covariances for binary diseases requires the use of multivariate liability threshold

model to account for both the binary nature of disease outcomes and the sample ascertainment associated with case control studies. In the univariate case, the linear mixed model can be viewed as an approximate form of the liability threshold model when individual relatedness is low [58–60]. Consequently, SNP heritability estimates can be obtained for disease outcomes by using standard MoM algorithms, where we can treat the binary outcome as a continuous outcome, fit a univariate linear mixed model to estimate heritability on the observed scale, and finally perform a scaling transformation to obtain heritability on the liability scale [58–60]. A recent study suggests that mvLMM can be similarly viewed as an approximated form of the multivariate liability threshold model [58]. Consequently, GECKO may be extended to estimate genetic and environmental covariances in case-control studies. However, as it is well recognized that the REML approach in the univariate linear mixed model can greatly underestimate the true heritability [58,59], it would be important to investigate whether the composite likelihood approach would produce unbiased covariance estimates as likelihood based approaches.

While we have focused on estimating both the genetic and environmental covariances in the present study, we acknowledge that the estimation of the environmental covariance is likely more susceptible to technical artifacts than the estimation of the genetic covariance. Specifically, technical artifacts such as population stratification or other confounding factors are likely absorbed into the second covariance term in the marginal likelihood in Eq (4) but not the first covariance term there. Consequently, such artifacts may influence the estimation of the environmental covariance which is also in the second term, while leaving alone the genetic covariance which is primarily in the first term. In the univariate LMM case, LDSC introduces an additional intercept parameter to control for population stratification in heritability estimation [10]. We have used an alternative strategy, relying on the fact that population stratification increases SNP-SNP correlations on top of what would be expected under LD alone, to extend GECKO to control for population stratification in covariance estimation (S1 Text). In particular, the extension of GECKO introduces three additional intercept parameters to control for population stratification that are present in the two unique samples of the two studies as well as in the common samples shared between the two studies, respectively. Such extension of GECKO effectively includes the univariate LDSC approach for controlling for population stratification as a special case. Unfortunately, the environmental covariance cannot be directly estimated in the extended GECKO due to model identifiability issues, unless we make an additional modeling assumption that the same level of population stratification occurs in the unique and common samples. In this case, only one intercept is needed to control for population stratification, allowing us to solve the model identifiability issue and estimate the environmental covariance. Examining the effectiveness of such extension in GECKO to control for population stratification is an important future research avenue.

## Supporting information

**S1 Fig. Comparison of genetic and environmental covariance estimation of different methods in simulations.** Compared methods include GECKO (purple), GNOVA (grey), and LDSC (green). Results are shown for the one study design with positive genetic or environmental covariance (first row: A, B, E, F) and the one study design with negative genetic or environmental covariance (second row: C, D, G, H). Boxplots display estimated genetic covariances (A, C) and environmental covariances (E, G) on y-axis versus the true covariances on x-axis across simulation replicates. Estimation accuracy is measured by the ratio of mean square errors (MSE), which contrast the MSE from GNOVA or LDSC with respect to GECKO, across various true covariances on x-axis, for genetic (B, D) and environmental covariances (F, H).

An MSE ratio below one suggests that GECKO performs worse than the other method; above one otherwise.
(TIF)

**S2 Fig. Comparison of genetic and environmental correlation estimation of different methods in simulations.** Compared methods include GECKO (purple), GNOVA (grey), and LDSC (green). Results are shown for the one study design (first row: A, B, G, H), two partially overlapped study design (second row: C, D, I, J), and two separate study design (third row: E, F). Boxplots display estimated genetic correlation (A, C, E) and environmental correlation (G, I) on y-axis versus the true covariances on x-axis across simulation replicates. Estimation accuracy is measured by the ratio of mean square errors (MSE), which contrast the MSE from GNOVA or LDSC with respect to GECKO, across various true correlation on x-axis, for genetic (B, D, F) and environmental (H, J) correlation. An MSE ratio below one suggests that GECKO performs worse than the other method; above one otherwise.
(TIF)

**S3 Fig. Comparison of genetic and environmental correlation estimation of different methods in simulations.** Compared methods include GECKO (purple), GNOVA (grey), and LDSC (green). Results are shown for the one study design with positive genetic or environmental correlation (first row: A, B, E, F) and the one study design with negative genetic or environmental correlation (second row: C, D, G, H). Boxplots display estimated genetic correlation (A, C) and environmental correlation (E, G) on y-axis versus the true correlation on x-axis across simulation replicates. Estimation accuracy is measured by the ratio of mean square errors (MSE), which contrast the MSE from GNOVA or LDSC with respect to GECKO, across various true correlation on x-axis, for genetic (B, D) and environmental correlation (F, H). An MSE ratio below one suggests that GECKO performs worse than the other method; above one otherwise.
(TIF)

**S4 Fig. Comparison of genetic and environmental covariance estimation of different methods in simulations.** Compared methods include GECKO (purple), GNOVA (grey), LDSC (green), and mvlmm(blue). Results are shown for the one study design (first row: A, B, G, H), two partially overlapped study design (second row: C, D, I, J), and two separate study design (third row: E, F). Boxplots display estimated genetic covariances (A, C, E) and environmental covariances (G, I) on y-axis versus the true covariances on x-axis across simulation replicates. Estimation accuracy is measured by the ratio of mean square errors (MSE), which contrast the MSE from GNOVA, LDSC or mvlmm with respect to GECKO, across various true covariances on x-axis, for genetic (B, D, F) and environmental covariances (H, J). An MSE ratio below one suggests that GECKO performs worse than the other method; above one otherwise.
(TIF)

**S5 Fig. Power comparison of different methods in detecting non-zero genetic and environmental covariances in simulations with functional annotations.** Compared methods include GECKO (purple, solid line), GNOVA (grey, dotted line).Results are shown for the one study design (first row: A, D), two partially overlapped study design (second row: B, E) and two separate study design (third row: C). Power (y-axis) of different methods are shown with respect to the true covariances (x-axis) for detecting non-zero genetic covariances (A, B, C) and environmental covariances (D, E). Power is shown based on a type I error of 0.05 but not a nominal p-value of 0.05. The power of GNOVA for detecting non-zero environmental covariance is not shown because GNOVA cannot test for environmental covariance.
(TIF)

**S6 Fig. Comparison of genetic and environmental covariance estimation of different methods in simulations with dense SNPs.** Compared methods include GECKO (purple), GNOVA (grey), and LDSC (green). Results are shown for the one study design (first row: A, B, G, H), two partially overlapped study design (second row: C, D, I, J), and two separate study design (third row: E, F). Boxplots display estimated genetic covariances (A, C, E) and environmental covariances (G, I) on y-axis versus the true covariances on x-axis across simulation replicates. Estimation accuracy is measured by the ratio of mean square errors (MSE), which contrast the MSE from GNOVA or LDSC with respect to GECKO, across various true covariances on x-axis, for genetic (B, D, F) and environmental covariances (H, J). An MSE ratio below one suggests that GECKO performs worse than the other method; above one otherwise.
(TIF)

**S7 Fig. Comparison of genetic and environmental covariance estimation of different methods in simulations with smaller heritability.** Compared methods include GECKO (purple), GNOVA (grey), and LDSC (green). Results are shown for the one study design (first row: A, B, G, H), two partially overlapped study design (second row: C, D, I, J), and two separate study design (third row: E, F). Boxplots display estimated genetic covariances (A, C, E) and environmental covariances (G, I) on y-axis versus the true covariances on x-axis across simulation replicates. Estimation accuracy is measured by the ratio of mean square errors (MSE), which contrast the MSE from GNOVA or LDSC with respect to GECKO, across various true covariances on x-axis, for genetic (B, D, F) and environmental covariances (H, J). An MSE ratio below one suggests that GECKO performs worse than the other method; above one otherwise.
(TIF)

**S8 Fig. Comparison of genetic and environmental covariance estimation of different methods in simulations using LD score from 1000 Genome reference panel.** Compared methods include GECKO (purple), GNOVA (grey), and LDSC (green). Results are shown for the one study design (first row: A, B, G, H), two partially overlapped study design (second row: C, D, I, J), and two separate study design (third row: E, F). Boxplots display estimated genetic covariances (A, C, E) and environmental covariances (G, I) on y-axis versus the true covariances on x-axis across simulation replicates. Estimation accuracy is measured by the ratio of mean square errors (MSE), which contrast the MSE from GNOVA or LDSC with respect to GECKO, across various true covariances on x-axis, for genetic (B, D, F) and environmental covariances (H, J). An MSE ratio below one suggests that GECKO performs worse than the other method; above one otherwise.
(TIF)

**S9 Fig. Comparison of genetic covariance estimation under mis-specified number of overlapped samples.** Compared methods include GECKO (purple), GNOVA (grey), and LDSC (green). Results are shown for $n_s$ being 250 (A, B), 500 (C, D), 1000 (E, F), 1250 (G, H), 1500 (I, J); Boxplots display estimated genetic covariances (A, C, E, G, I) on y-axis versus the true covariances on x-axis across simulation replicates. Estimation accuracy is measured by the ratio of mean square errors (MSE), which contrast the MSE from GNOVA or LDSC with respect to GECKO, across various true covariances on x-axis, for genetic covariances (B, D, F, H, J). An MSE ratio below one suggests that GECKO performs worse than the other method; above one otherwise.
(TIF)

**S10 Fig.** Comparison of environmental covariance estimation under mis-specified number of overlapped samples. Compared methods include GECKO (purple), GNOVA (grey), and LDSC (green). Results are shown for $n_s$ being 250 (A, B), 500 (C, D), 1000 (E, F), 1250 (G, H),

1500 (I, J); Boxplots display estimated environmental covariances (A, C, E, G, I) on y-axis versus the true covariances on x-axis across simulation replicates. Estimation accuracy is measured by the ratio of mean square errors (MSE), which contrast the MSE from GNOVA or LDSC with respect to GECKO, across various true covariances on x-axis, for environmental covariances (B, D, F, H, J). An MSE ratio below one suggests that GECKO performs worse than the other method; above one otherwise.
(TIF)

**S11 Fig. Genetic correlation estimates from LDSC and GECKO for pairs of 22 human complex traits.** The upper triangular represents the genetic correlation estimates by LDSC while the lower triangular respresents the estimates by GECKO. The cross in the square represents the significant genetic correlation after Bonferroni Correlation.
(TIF)

**S1 Table. Information for the summary statistics of 22 traits from 5 GWAS studies.** The table lists the phenotype name, category, abbreviation, number of individuals, reference, and downloaded websites for each of the 22 traits.
(XLSX)

**S2 Table. Genetic correlation estimates by GECKO and LDSC.** The table lists the phenotype name abbreviation of the first study, phenotype name abbreviation of the second study, genetic covariance, heritability of the first trait, heritability of the second trait, genetic correlation estimates and corresponding p-value of GECKO, LDSC and GNOVA.
(XLSX)

**S3 Table. Environmental correlation estimates by GECKO.** The table lists the phenotype name abbreviation of the first study, second study, environmental covariance estimates, environmental variance of the first trait, environmental variance of the second trait, environmental correlation estimate, corresponding p-value, and supporting information.
(XLSX)

**S4 Table. Genetic and environmental covariance estimate under dense SPNs set by GECKO.** The table lists the phenotype name abbreviation of the first study, second study, genetic and environmental covariance and correlation estimate of GECKO and corresponding p-value.
(XLSX)

**S1 Text. Supplementary text for the methods.**
(DOCX)

## Author Contributions

**Conceptualization:** Xiang Zhou.

**Data curation:** Boran Gao.

**Formal analysis:** Boran Gao.

**Funding acquisition:** Xiang Zhou.

**Investigation:** Boran Gao.

**Methodology:** Boran Gao, Can Yang, Jin Liu, Xiang Zhou.

**Project administration:** Xiang Zhou.

**Resources:** Xiang Zhou.

**Software:** Boran Gao.

**Supervision:** Xiang Zhou.

**Validation:** Boran Gao.

**Visualization:** Boran Gao.

**Writing – original draft:** Boran Gao.

**Writing – review & editing:** Boran Gao, Can Yang, Jin Liu, Xiang Zhou.

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
