## [Decision Letter · Decision Letter 0]

10 Aug 2020

Dear Dr Gao,

Thank you very much for submitting your Research Article entitled 'Accurate genetic and environmental covariance estimation with composite likelihood in genome-wide association studies' to PLOS Genetics. Your manuscript was fully evaluated at the editorial level and by independent peer reviewers. The reviewers appreciated the attention to an important problem, but raised some substantial concerns about the current manuscript. Based on the reviews, we will not be able to accept this version of the manuscript, but we would be willing to review again a much-revised version. We cannot, of course, promise publication at that time.

If you decide to revise the manuscript for further consideration at PLOS Genetics, please aim to resubmit within the next 60 days, unless it will take extra time to address the concerns of the reviewers, in which case we would appreciate an expected resubmission date by email to plosgenetics@plos.org.

[LINK]

We are sorry that we cannot be more positive about your manuscript at this stage. Please do not hesitate to contact us if you have any concerns or questions.

Yours sincerely,

Michael P. Epstein

Associate Editor

PLOS Genetics

Hua Tang

Section Editor: Natural Variation

PLOS Genetics

Reviewer's Responses to Questions

**Comments to the Authors:**

Reviewer #1: This is a great paper with contributions to an important problem. Overall, the manuscript is well-structured and clearly written. Although I have some concerns and questions about the technical details of GECKO and some analyses presented in the paper, all my comments should be addressable in principle.

Major comments:

1. It was not immediately obvious to me if GECKO assumes the design matrix (X) to be fixed or random. Methods for genetic correlation estimation (and frankly, heritability estimation as well) have not been consistent on this issue. My recollection is that the GREML approach and GNOVA assume a fixed design while LDSC is based on a random design. While the field seems to have accepted that genetic correlation estimation is robust to these two different designs (at least empirically), whether this will affect the estimation of environmental covariance is unclear. Under a random design, the intercept of LDSC is the phenotypic covariance of two traits. Under a fixed design, I am under the impression that the intercept may actually be the environmental covariance instead. When the goal is to estimate the genetic covariance, such differences does not matter since the slope in LDSC is always a good estimator for rho_g. But the interpretation of intercept is completely different under two model assumptions. It will be important to clarify the assumptions in GECKO and provide some discussions on whether the environmental covariance is robust to different assumptions.

2. The field has not paid as much attention to the environmental covariance compared to genetic covariance. One reason is that the environmental covariance is more susceptible to technical artifacts (such as insufficiently controlled confounding) in the GWAS summary statistics. In fact, this was a highlight in genetic covariance estimation since if there are any technical correlations between two sets of GWAS, such correlations will most likely be independent from LD. The first LDSC paper on contrasting confounding from polygenicity is a good example on this issue (in the single-trait context). Suppose there is unadjusted population stratification in a GWAS, such inflation is randomly applied to SNPs in the genome while true polygenic effects are stronger in genomic regions with stronger LD. Therefore, all the technical bias will be absorbed into the intercept term of single-trait LDSC. The multi-trait extension of this problem is that technical correlation of two traits caused by population stratification will show up in the intercept of LDSC. I have been using LDSC to illustrate the problem since it's a famous method but I don't think GECKO is immune to the exact same issue. If you run two GWASs without adjusting for PCs, and apply GECKO to their sumstats, I expect the "environmental covariance" estimates to be severely biased while the genetic covariance estimates will remain solid. If the authors can come up with an approach to address this problem, it will improve this manuscript. Otherwise, at least treat this as a serious problem and list it as a limitation.

3. Only ~600k SNPs were used in the real data analyses which seems low. I wonder whether the somewhat heuristic composite likelihood approach in GECKO is still effective when there are dense SNPs with extensive LD in the analysis. This is important considering that both simulation and real data analysis in this paper were based on relatively low numbers of SNPs. LDSC also requires trimming of sumstats by taking the overlap with HAPMAP SNPs while GNOVA uses all available SNPs. After all, if there are few SNPs with minimal LD in the data, genetic and environmental covariance estimation becomes a much simpler problem. The tricky issue really is LD here.

4. It is helpful that GECKO can partition genetic covariance by annotation. It will be helpful if the authors can provide some discussions on whether the procedure can be easily generalized to overlapping annotations.

5. The inflated type-I error of GNOVA illustrated in Table 1 is striking and also inconsistent with the original publication of GNOVA. However, these results are also inconsistent with what is shown in Figure 2. There, when the true genetic covariance is 0, all three approaches seem to have comparable type-I error rates. Why the contradiction?

Minor comments:

6. Figure 2's panels are labeled differently between the main text and the actual plot. See line 403 on page 24. I think the authors meant Fig2D and Fig2E instead of panels B and D.

7. In the real data analysis, is there any trait pair showing significant genetic and environmental covariance but with opposite signs? How to interest these results?

8. Fig1E seems to suggest that LDSC had noisy estimates on genetic covariance but it contradicts Fig1F.

9. In the simulation corresponding to Fig2C, if we apply GECKO, should we expect to see near-null results with a "power" of 5%? In practice, it may not be obvious if two GWASs have overlapping samples. Consequently, it may not be easy (or even possible) to distinguish a lack of environmental covariance and a lack of sample overlap. It may be helpful to add some discussions on this issue.

10. There have been some recent developments on genetic covariance estimation. For example, HDL (https://www.nature.com/articles/s41588-020-0653-y) uses a likelihood-based approach to improve the efficiency of estimation and SUPERGNOVA (https://www.biorxiv.org/content/10.1101/2020.05.08.084475v1.abstract) de-correlates the summary statistics first and then apply standard regression analysis. In addition, trans-ethnic estimation is an important topic and a clear future direction (https://www.biorxiv.org/content/10.1101/803452v3). Since these are very recent papers, it wouldn't be fair to ask for comparisons between GECKO and these methods. But some discussions regarding these new developments will improve the manuscript.

Reviewer #2: Review: Accurate genetic and environmental covariance estimation with composite likelihood in genome-wide association studies

Summary:

Gao et al introduce a novel statistical approach, GECKO, to estimate genetic and environmental covariance between complex traits using only GWAS summary statistics with a computationally efficient composite likelihood estimator. The approach is motivated by the lack of likelihood-based approaches in summary-statistic-based genetic covariance estimation (e.g., LDSC, GNOVA both use MoM estimators). The authors performed extensive simulations to validate their approach and demonstrate its improved performance. They then followed with application to real data as an illustration. The idea is interesting, and I found the manuscript extremely well written and straightforward to follow. I detail my main comments below.

Major Comments:

1. The authors performed a considerable number of simulations demonstrating the mostly improved performance of GECKO with respect to LDSC and GNOVA. However, all simulations were performed essentially under the correct generative model, with no inspection of performance under model misspecification. A few scenarios come to mind:

a) LD pattens match exactly between the focal population and LD score computation. To investigate this point further, can the authors showcase the performance of all three methods when 1000G LD information is used to compute LD scores?

b) How does GECKO (and LDSC, GNOVA) perform under misspecification of n_s?

c) How does stratification affect the performance of GECKO? A straightforward simulation would be to generate both phenotypes as a function of several leading PCs, in addition to the remaining genetic + environmental model.

Minor Comments:

1. MoM estimators are unbiased, and it is not unexpected to see estimates largely agree across all three methods. However, as the authors note, GECKO’s development was partly motivated by improving statistical power. The use of MSE ratios is interesting to get a relative sense of performance, but still find it difficult to judge what my expectation should be here (e.g., Is 1.05 really all that different). I appreciated the authors decision to include power calculations in the non-stratified setting given its clear interpretation but noticed their absence in the functional stratified simulations.

2. Replace “*” with “x” in estimates/p-values.

Reviewer #3: Gao et al. presented GECKO, a statistical method that aims to estimate both genetic and environmental covariances between different traits using GWAS summary statistics. The key idea of GECKO is to approximate complicated joint likelihood with a relatively simple composite likelihood, which holds the potential to be computationally fast if the weight parameters in the composite likelihood are correctly specified. The authors evaluated the performance of GECKO by simulations as well as the analysis of 22 traits collected from five large scale GWASs.

I have the following concerns about this paper.

Main comments:

1. Simulations: While the authors have simulated >200 scenarios, these mostly reflect the small step size used to evaluate the impact of genetic covariance (-0.4 to 0.4) or environmental covariance (-0.4 to 0.4), whereas the other key parameter, e.g., heritability, which is also extremely important, was set at constant values. The values used in the simulations do not seem to reflect those seen in real data that the authors showed later on. Overall, I feel that the current parameter values used in simulations are too different from what one would expect from real data, which make it difficult to judge whether these simulation results are useful. To get a better sense on the performance of GECKO, I think it is important to simulate data that are more representative of real data. I would suggest the authors to do the following:

a) Estimate heritability for the 22 traits collected from real GWASs. Estimate genetic covariances and environmental covariances using these real data. Use these estimates as a guideline to determine the range of parameter values for simulations.

b) The data were simulated from the true model. What if the model is mis-specified? In particular, what if the weights are not optimal? The authors should illustrate that GECKO is robust to misspecification of the model and weights.

c) Evaluation of type I error at 0.05 level seems to be too liberal. I am curious to see if GECKO still has controlled type I error rate at a more stringent significance level, e.g. 0.005.

2. Real data analysis:

a) Please show the heritability for the 22 traits. This will help readers interpret the results and connect with simulations.

b) I am surprised that all estimated covariances are very small. However, even for such small covariances, they are highly significant. For example, the positive genetic covariance between BMI and WBC (covariance = 7.91*10-3; p value =2.24*10-12). What is the biological significance of such a small but highly significant covariance? Can you also show estimation error of the estimated covariances? What I am surprised is that all of the estimated covariances are extremely small. I am not sure that these results are biologically significant. The authors should provide better interpretation of the results and show that these estimates are reliable.

c) Since 223 pairs of traits were tested, multiple testing issue should be considered. I am also curious to see the performance of GEKCO at a more stringent significance threshold. Considering correlation among the 223 pairs, 100 independent tests might be a good start. With this, using Bonferroni, the threshold will be 0.05/100=0.0005.

3. The methods section should include more details on how the weights are estimated as this is the key of GECKO.

4. In terms of comparison, shouldn’t you also compare with mvLMM?

5. Can you comment on the computing time of GECKO? How does it compare with other compared methods with respect to computing speed?

6. Can you comment on how to use GECKO for admixed populations?

Minor comments:

1. Line 404: How did you compare the power of different methods at fixed type I error rate?

2. Can you comment on why the relative powers of different methods are different across study designs? What is the intuition behind this difference?

3. For real data analysis, it would still be interesting to include GNOVA’s estimates for covariance.

**Have all data underlying the figures and results presented in the manuscript been provided?**

Reviewer #1: Yes

Reviewer #2: Yes

Reviewer #3: Yes

PLOS authors have the option to publish the peer review history of their article (what does this mean?). If published, this will include your full peer review and any attached files.

Reviewer #1: No

Reviewer #2: No

Reviewer #3: No

---

## [Decision Letter · Decision Letter 1]

2 Dec 2020

Dear Dr Gao,

We are pleased to inform you that your manuscript entitled "Accurate genetic and environmental covariance estimation with composite likelihood in genome-wide association studies" has been editorially accepted for publication in PLOS Genetics. Congratulations!

Yours sincerely,

Michael P. Epstein

Associate Editor

PLOS Genetics

Hua Tang

Section Editor: Natural Variation

PLOS Genetics

Comments from the reviewers (if applicable):

Reviewer's Responses to Questions

**Comments to the Authors:**

Reviewer #1: The authors did an excellent job revising the manuscript and performing new analyses. All my previous comments have been sufficiently addressed and I have no additional comment or concern.

Reviewer #2: The authors have addressed all my comments.

Reviewer #3: The authors have appropriately addressed my previous concerns. I don't have any additional comments.

**Have all data underlying the figures and results presented in the manuscript been provided?**

Reviewer #1: Yes

Reviewer #2: Yes

Reviewer #3: Yes

PLOS authors have the option to publish the peer review history of their article (what does this mean?). If published, this will include your full peer review and any attached files.

Reviewer #1: No

Reviewer #2: No

Reviewer #3: No

**Data Deposition**

http://datadryad.org/submit?journalID=pgenetics&manu=PGENETICS-D-20-00904R1

**Press Queries**

---

## [Editor Report · Acceptance letter]

29 Dec 2020

PGENETICS-D-20-00904R1 

Accurate genetic and environmental covariance estimation with composite likelihood in genome-wide association studies 

Dear Dr Gao, 

We are pleased to inform you that your manuscript entitled "Accurate genetic and environmental covariance estimation with composite likelihood in genome-wide association studies" has been formally accepted for publication in PLOS Genetics! Your manuscript is now with our production department and you will be notified of the publication date in due course.

With kind regards,

Melanie Wincott

PLOS Genetics

On behalf of:
